# Antigen-Heterologous Vaccination Regimen Triggers Alternate Antibody Targeting in SARS-CoV-2-DNA-Vaccinated Mice

**DOI:** 10.3390/vaccines12030218

**Published:** 2024-02-20

**Authors:** Anders Frische, Karen Angeliki Krogfelt, Anders Fomsgaard, Ria Lassaunière

**Affiliations:** 1Department of Virus & Microbiological Special Diagnostics, Statens Serum Institut, 2300 Copenhagen, Denmark; afi@ssi.dk (A.F.); karenak@ruc.dk (K.A.K.); afo@ssi.dk (A.F.); 2Section of Molecular and Medicinal Biology, Department of Science and Environment, Roskilde University, 4000 Roskilde, Denmark; 3PandemiX, Center for Interdisciplinary Study of Pandemic Signatures, Institute for Science and Environment, Roskilde University, 4000 Roskilde, Denmark; 4Infectious Diseases Unit, Clinical Institute, University of Southern Denmark, 5230 Odense, Denmark

**Keywords:** SARS-CoV-2, epitope mapping, microarray, neutralizing antibodies, DNA vaccine

## Abstract

An in-depth analysis of antibody epitopes following vaccination with different regimens provides important insight for developing future vaccine strategies. B-cell epitopes conserved across virus variants may be ideal targets for vaccine-induced antibodies and therapeutic drugs. However, challenges lie in identifying these key antigenic regions, and directing the immune system to target them. We previously evaluated the immunogenicity of two candidate DNA vaccines encoding the unmodified spike protein of either the SARS-CoV-2 Index strain or the Beta variant of concern (VOC). As a follow-on study, we characterized here the antibody binding profiles of three groups of mice immunized with either the DNA vaccine encoding the SARS-CoV-2 Index strain spike protein only, the Beta VOC spike protein only, or a combination of both as an antigen-heterologous prime-boost regimen. The latter induced an antibody response targeting overlapping regions that were observed for the individual vaccines but with additional high levels of antibody directed against epitopes in the SD2 region and the HR2 region. These heterologous-vaccinated animals displayed improved neutralization breadth. We believe that a broad-focused vaccine regimen increases neutralization breadth, and that the in-depth analysis of B-cell epitope targeting used in this study can be applied in future vaccine research.

## 1. Introduction

In December 2019, the SARS-CoV-2 Index strain emerged in Wuhan, China. Since then, the virus has evolved substantially, acquiring consecutively more amino acid changes in the major surface glycoprotein, the spike protein, which is the primary target for vaccine- and infection-induced neutralizing antibodies. Following the introduction of SARS-CoV-2 into Europe, the ancestral strain with the spike D614G substitution became dominant, followed by the emergence of a myriad of variants that include Alpha, Beta, Gamma, Delta, and several Omicron variants and recombinants. Immune pressure likely contributed to virus evolution that altered antigenicity, leading to escape of vaccine-induced immunity, natural immunity, and hybrid immunity. The pandemic has seen to an unprecedented and ongoing race to develop novel and updated vaccines that generate effective broadly neutralizing antibodies that are able to protect from severe disease against an everchanging virus.

SARS-CoV-2 vaccine development focused primarily on the spike protein due to its essential functions in virus entry, which makes it an ideal target for protective antibody responses. The spike protein is a trimer, and each monomer is built up by two non-covalently associated subunits S1 and S2. SARS-CoV-2 enters host cells by binding of the viral receptor-binding motif (RBM), located in the receptor-binding domain (RBD) of the spike protein, to the host cell receptor, angiotensin-converting enzyme 2 (ACE2). The RBD embedded in the S1 subunit shifts between a shielded ‘down’ conformation and an “open” receptor-binding conformation. Pre-cleavage at the S1/S2 cleavage site at the junction between S1 and S2 by furin or other cellular proteases promotes the RBD ‘up’ conformation, priming ACE2 binding [1,2,3]. In the ‘up’ position, the RBD can bind to the ACE2 receptor, leading to conformational changes in the spike and exposure of the S2’ cleavage site. To facilitate membrane fusion, S2’ is cleaved either by cell surface proteases such as TMPRSS2 or furin, or by cathepsin in the endosomal pathway, leading to a destabilization of the pre-fusion trimer and detachment of S1. Mediated by the S2 domain of the spike protein, the viral envelope fuses with the cell membrane through the insertion of the fusion peptide and the formation of a six-helix bundle by heptad repeat 1 and 2, pulling the two membranes together and resulting in fusion. 

Rational vaccine design or monoclonal antibody therapy relies on knowledge about B-cell epitopes generating effective broadly neutralizing antibodies. As a host receptor engaging viral protein domain, the RBD is the primary target for neutralizing antibodies and a clear target for intervention strategies [4,5,6]. RBD-specific antibodies directed against ACE2 binding epitopes are classified as class 1 or 2 antibodies [7]. Class 1 antibodies bind only RBD ‘up’ epitopes and class 2 antibodies bind both RBD ‘up’ and ‘down’ epitopes. Conversely, RBD-specific antibodies of classes 3 and 4 are non-ACE2 blocking antibodies that bind ‘up’ and ‘down’ (class 3) or only ‘up’ (class 4). Multiple factors contribute to an enhanced frequency of mutations in the RBD, thereby influencing vaccine immunogenicity and monoclonal antibody recognition [8]. The N-terminal domain (NTD) upstream of the RBD in the S1 subunit, similarly known to elicit potent neutralizing antibodies [9,10,11], is also a region of highly mutated residues primarily centered around a supersite located in the N3 loop (residues 141–156) [9,11]. Deletions compromising neutralizing antibody efficiency are known to exclusively affect the S1 regions outside the RBD [12,13]. The key to effective broadly neutralizing antibodies is therefore to target epitopes that are conserved across variants. Protein function and structural constraints are believed to have a strong impact on the likeliness that a gene sequence will mutate [14]. Targeting conformationally “locked” regions could be crucial for effective neutralization.

Statens Serum Institut, Denmark, developed two candidate SARS-CoV-2 vaccines that encode either the unmodified spike protein of the Index strain or the spike protein of the Beta VOC (PANGO lineage B.1.351) [15,16]. Both vaccines are immunogenic in rabbits and mice, producing robust binding and neutralizing antibody responses. In studies conducted by Lassauniere et al. [15], where CB6F1 mice received a homologous regimen of three immunizations of either of these vaccines or a combination of both, the authors observed an increased breadth of neutralization responses when the vaccines were mixed. In this study, we aim to determine if this increased breadth is the result of different epitopes being targeted. To this end, we study serum from mice that received homologous or heterologous vaccine regimens. We use microarray technology with circular constrained peptides of the entire Index strain spike ectodomain to identify targeted B-cell epitopes.

## 2. Materials and Methods

### 2.1. Study Population

The DNA vaccines, vaccination strategies, and animal experiments are described in detail elsewhere [15,16]. In brief, eight-week-old female CB6F1 mice (Envigo, Horst, The Netherlands) were immunized with 50 µg of a DNA plasmid vector, encoding either the complete and unaltered spike protein of the SARS-CoV-2 Index strain (pNTC-Spike), derived from the Wuhan-Hu-1 strain (MN908947), or of the Beta (B.1.351) strain (pNTC-Spike.351). Nature Technology Corporation (Lincoln, NE, USA) subcloned human codon-optimized SARS-CoV-2 spike sequences synthesized by GeneArt (Thermo Fisher Scientific, Dreieich, Germany) into the NTC8685-eRNA41H vector backbone and produced the vaccines in NTC4862 E. coli cells (DH5α attλ::P5/6 6/6-RNA-IN-SacV, Cmr) using their antibiotic-free RNA-OUT selection procedure. The vaccine stocks were provided in a concentration of 10 mg/mL in phosphate-buffered saline (PBS) [15]. Immunizations were performed in weeks 0, 2, and 4. Groups of five mice were immunized intradermally with needle injection at the base of the tail according to three different vaccine regimens (Figure 1). The first group received only the pNTC-Spike vaccine at weeks 0, 2, and 4, while the second group received only the pNTC-Spike.351 vaccine at the same time points. These antigen-homologous groups are hereafter referred to as the Index and Beta groups, respectively. The third group received a combination of the two vaccines as the pNTC-Spike vaccine administered at weeks 0 and 2, and the pNTC-Spike.351 vaccine administered at week 4. The latter antigen-heterologous group is hereafter referred to as the mixed group. Blood samples were taken two weeks after the final vaccination. The animals were housed in facilities at Statens Serum Institut, Copenhagen, Denmark. All procedures were supervised by the laboratory animal veterinarians and complied with the Danish legislation, which is based on EU Directive 2010/63/EU on the protection of animals used for scientific purposes. The experiments received ethical approval from The Animal Experimentation Council, the National Competent Authority within this field (approval number 2017-15-0201-01322).

### 2.2. Microarray Method

Epitope mapping was performed on pooled sera from each group of mice, according to vaccination regimen, using a custom-made microarray with 10′mer cyclic overlapping peptides spanning the full Index strain spike protein and 35 additional mutation sequences representing key mutations throughout the spike protein. To perform the epitope mapping, the peptide microarray obtained from PEPperPRINT (Heidelberg, Germany) was adjusted to room temperature and assembled in accordance with the manufacturer’s instructions. Subsequently, a washing buffer comprising Dulbecco’s phosphate-buffered saline (DPBS) with 0.05% Tween20, at a pH of 7.4, was added to each subarray. After incubation at room temperature for 15 min, the washing buffer was removed using a pipette and each subarray was blocked in 400 µL of Rockland Blocking Buffer (MB-070, Rockland Immunochemicals, Pottstown, PA, USA) at room temperature for 30 min. To evaluate non-specific reactions, we then pre-treated the peptide array with a secondary antibody/streptavidin solution diluted in staining buffer (DPBS with 0.005% Tween20 and 10% blocking buffer, pH 7.4) as described below and scanned. The microarray was then reinserted into the PEPperPRINT microarray cassette and incubated with staining buffer for 15 min to equilibrate. The staining buffer was then removed with a pipette. Diluted in staining buffer, 400 µL of each serum pool was pipetted into separate subarrays, and the microarray was left overnight at 2 to 8 °C on an orbital shaker at 140 rpm and in the dark. The following day, sample dilutions were removed by pipetting, followed by a two-times wash with 400 µL wash buffer for 10 s on an orbital shaker at 140 rpm, protected from light. Moreover, 400 µL of biotinylated goat anti-mouse IgG F(c) (cat. #31805, Invitrogen, Thermo Fisher Scientific, Waltham, MA, USA) was added to each subarray. This secondary antibody was diluted 1:500 in staining buffer and the microarray was subsequently incubated for 45 min at room temperature. The secondary antibody was removed by pipetting, followed by 2× wash with 400 µL wash buffer as previously described, and 400 µL streptavidin Alexa Fluor 647 conjugate (Thermo Fisher Scientific, Waltham, MA, USA) diluted 1:750 in staining buffer was added, and the peptide microarray was incubated at room temperature for 45 min. Each subarray was then washed two times with 400 µL wash buffer as previously described, and the entire array slide was submerged two times into dipping buffer (1 mM Tris buffer, pH 7.4), followed by a 1 min centrifugation at 1000 rpm to completely remove all dipping buffer. Microarray was scanned on a microarray scanner (SureScan from Agilent, Santa Clara, CA, USA) and analyzed using MAPIX Analyzer Software v. 9.1.0. The microarray was then re-inserted into the microarray cassette and equilibrated as previously described. After removal of the staining buffer using a pipette, the microarray was finally incubated with an anti-HA PEPperCHIP^®^ control antibody from PEPperPRINT diluted 1:2000 in staining buffer and incubated at room temperature for 45 min. Subarrays were washed twice with 400 µL wash buffer, and, after disassembly, the entire array slide was dipped twice into dipping buffer, centrifuged, scanned, and analyzed as previously described. All RT incubations were carried out in the dark using an orbital shaker at 140 rpm. 

### 2.3. Statistical Analysis and Calculations

Fluorescence (relative fluorescence units, RFU) was determined as fluorescence intensity at 635 nm with a subtraction of background determined at 635 nm from an area manually set within each subarray in MAPIX. Microarray fluorescence data were imported into Excel. Fluorescence was calculated as the mean of Δ-fluorescence from two identical peptides and normalized according to the mean of a positive HA control (*n* = 48) included on each subarray. Targeted epitopes were defined as the fluorescence intensities above the cut off. The cut off was defined as
$$Cut\;off=\bar{X}+SD\times f$$
where $\bar{X}$ is the mean of *n* negative control determinations (*n* = 96) included on each subarray, SD is the standard deviation, and f is the standard deviation multiplier with a confidence interval of 99.0% as provided by Frey et al. [17].

## 3. Results

### 3.1. Selection of Animals

To characterize the antibody binding profiles in DNA-vaccinated mice, we selected animals grouped according to the three different immunization regimens described in Section 2. Displaying quite distinct cross-neutralization profiles (Appendix A) [15] and immunized with three different vaccine regimens, we hypothesized that these animals would target different epitopes on the spike protein. To study this and identify potentially broadly neutralizing epitopes, we assessed the overall antibody binding profiles from pools of serum from each group of animals having followed the same immunization regimen using microarray epitope mapping.

### 3.2. Antibody Binding Profiles

We designed a peptide microarray using cyclic 10′mer sequences, with eight overlapping amino acids and two amino acid shifts, spanning the full Index strain spike protein and 35 additional sequences of mutations representing important VOCs at the time of design. Because of a very limited amount of serum accessible for each animal, microarray analysis was carried out using the serum pools of the four to five individual mice included in each vaccination regimen group. The amount of individual mouse serum in each pool was normalized according to previously established levels of ELISA binding IgG titers (Appendix A), adding equal titers of spike-ectodomain-specific antibody to control for confounding variables, such as variable levels of overall spike IgG levels between animals. Each group of mice was analyzed in a 1:20 dilution and a 1:100 dilution, allowing assessment of low-affinity and high-affinity antibodies. Targeted (active) epitopes were defined as epitopes with relative fluorescence units (RFU) above a cut-off value calculated from a negative control set included on each subarray.

#### 3.2.1. Targeting of RBD Epitopes

We observed moderate targeting and binding within the entire RBD, with maximum RFU values of 8087, 8529, and 7203 (diluted 1:20) in the Index-vaccinated group, the Beta-vaccinated group, and the mixed-vaccinated group, respectively (Figure 2). Maximum RBD targeting was found within the RBM in the Beta-vaccinated group, indicating an ACE2 blocking focus (antibody classes 1 and 2), and just outside the RBM in the remaining two groups. The Index-vaccinated group and the mixed-vaccinated group had the apex RBD peak at peptide 429, spanning residues 420–429. This region overlaps with two known monoclonal class 4 antibodies, CR3022 and COVA1-16 [18,19]. The apex RBD peak in the Beta-vaccinated group was found at peptide 473, spanning residues 464–473. This peptide, also targeted by the other two groups, overlaps with class 1 mAb GAR05 epitope [20,21]. An adjacent epitope, spanning residues 478–501, overlapping with another class 1 mAb, CB6 [19,22], similarly shows a generally higher level of antibody targeting in all three groups. 

#### 3.2.2. Mutational Sensitivity

The ability to broadly neutralize different variants could be influenced by the capacity of antibodies to tolerate viral mutations. To test the animals in each vaccine regimen’s resilience towards mutations, we included peptides of 35 of the most prominent mutations at the time of design of the microarray, including neutralization-resistant K417N and receptor-binding motif substitution E484K. 

Substitutions C136F and D138Y and deletion DEL144 in the NTD supersite heavily affected binding in the Beta-vaccinated group and to a lesser extent the mixed-vaccinated group (Figure 3). In contrast, enhanced binding was observed in the Index-vaccinated group. These mutations resulted in a 58% reduction in total binding, from 188,843 RFU to 80,099 RFU, in the Beta-vaccinated group in serum diluted 1:20. In comparison, the total binding in the same region in the mixed-vaccinated group decreased 24% and increased 94% in the Index-vaccinated group. Although a moderate effect, E484K substitution leads to a general loss of antibody binding; in contrast, K417N results in a surprising increase in signal in all the groups. The N501Y substitution within the RBM, which is believed to enhance ACE2 binding affinity [23,24], does not seem to affect antibody binding in any of the groups. In contrast, Alpha variant substitution A570D in the SD1 region increased binding in the Index-vaccinated group and the Beta-vaccinated group, but it is nearly untargeted in the mixed-vaccinated group. The D614G substitution is generally low-targeted; however, the mixed-vaccinated group is the only group with enhanced targeting of the mutated form. Except for the D614G substitution, the groups vaccinated with the pNTC.Spike.351, in particular the Beta-vaccinated group, display enhanced antibody binding to all Beta mutations. The Index-vaccinated group, the group with the highest neutralizing antibody levels against the D614G variant and the Delta AY.4. variant, however, displays generally low binding to all the Beta mutation peptides and all the included Delta mutation peptides.

#### 3.2.3. Consistently Targeted Regions

To identify highly immunogenic regions on the spike protein targeted by DNA-vaccine-induced antibodies in mice, we evaluated the epitopes consistently targeted between the three groups. At a serum dilution of 1:20, all three groups consistently targeted multiple epitopes within the region spanning residues 130–179 around the N-terminal domain supersite, with maximum intensity at peptide 141 (EFQFCNDPFL) consisting of amino acids 132–141 (Figure 4). This site, along with the SD3 site spanning residues 1144–1153, displays moderate to very high targeting, with a minimum fluorescence exceeding 6000 RFU. The neighboring epitopes adjacent to these hotspots, spanning residues 148–163, 1138–1151, and 1199–1217, were similarly consistently targeted with low to high expression (low 1835 RFU; high 9117 RFU). The RBD site at peptide 429 (DYNYKLPDDF) spanning residues 420–429, and peptides 1263 and 1265 spanning the area of residues 1254–1265 in the connector domain in the periphery of the protruding spike, equally showed moderate binding levels (>6000 RFU) in all three groups. At a higher serum dilution of 1:100, peptides 141, 1263, and 1265 also displayed notable antibody binding, indicating high-affinity/high-abundance antibodies (Appendix A). 

#### 3.2.4. Non-Consistently Targeted Regions

In addition to the consistently targeted regions, the Index-vaccinated group and the Beta-vaccinated group both had high binding to peptide 579 (ADTTDAVRDP), spanning residues 570–579 in the SD1 region, with RFUs exceeding 12,000 (12,733, 12,081), and peptide 1187 (NIQKEIDRLN; RFU 6285, 8163) in the HR2 region in a 1:20 serum dilution. Diluted 1:100, peptide 579 is the apex peak in the Index-vaccinated group, with a 2-fold (1.9) higher antibody binding signal than peptide 1113 (Appendix A). These two peptides are targeted with equal fluorescence at a serum dilution of 1:20. This SD1 epitope is significantly less targeted in the mixed-vaccinated group (RFU = 2743). The Beta group displayed a markedly higher (3.6-fold) level of antibodies targeting NTD peptide 141 compared to the lowest level observed in the Index-vaccinated group, and a 4.2-fold higher level of the 1265 peptide (Figure 4B,C). Apart from the consistently targeted regions shared between all the groups, the mixed-vaccinated group showed unique high levels of antibody binding to peptides 625 and 627 spanning residues 616–627 very close to the 630-loop in the SD2 region, with a maximum RFU of 21,676 compared to the Index-vaccinated group and the Beta-vaccinated group, with low intensities of 1688 and 3404, respectively. Peptide 1169 (TSPDVDLGDI) in the HR2 region similarly showed very high antibody binding in the mixed-vaccinated group (RFU = 29,320) compared to 4283 and 6584 RFU as observed in the Index and in the Beta-vaccinated groups, respectively. The Beta-vaccinated group showed high general antibody levels covering most parts of the spike protein and focusing less on individual epitopes, in contrast to the more focused response observed for the mixed-vaccinated group (Figure 4C). The serum antibodies of the mixed-vaccinated group do, however, seem to target regions in the NTD, RBD, CR, and CH also observed for the Beta-vaccinated group, which were absent to low in the Index-vaccinated group (Figure 4C). Of note, none of the groups displayed noticeably high levels of binding to the S1/S2 cleavage site, the S2’ cleavage site, or the fusion peptide. 

## 4. Discussion 

This study applied peptide microarray technology to map B-cell epitopes targeted by SARS-CoV-2 spike-specific antibodies induced by three different regimens of DNA vaccines to aid rational vaccine development in the future. It is a follow-up study of the development and pre-clinical immunogenicity studies of the SARS-CoV-2 DNA vaccines described in Lassauniere et al. [15,16]. These studies found an enhanced breadth in heterologous-vaccinated mice that received a combination of a DNA plasmid encoding the full Index strain spike protein sequence and an identical plasmid vector encoding the full Beta VOC spike protein. Being one of the first variants bearing immune escape capabilities such as the E484K and K417N/T substitutions, the Beta VOC represented a highly antibody-resistant variant compared to the ancestral Index strain. To determine if the observed improved ability to cross-neutralize different VOCs in the heterologous-vaccinated mice is the result of different antibody targeting, we established B-cell epitope profiles from polyclonal serum from groups of mice representing each vaccination regimen. The animals following the antigen-heterologous vaccination regimen, combining both of the two used DNA vaccine constructs leading to a slightly improved cross-neutralizing capacity, showed the most distinct binding profile.

The ideal vaccine should induce a broad neutralizing antibody response, conferring protection against present and emerging virus strains. Our results indicate that using an antigen-homologous vaccine strategy directs the immune system into a focused strain-specific response. In contrast, applying an antigen-heterologous vaccination regime resulted in broader cross-neutralization [15]. We observed an antigen-heterologous vaccine regimen antibody profile targeting common regions in the NTD, RBD, and SD3 equally targeted by the antigen homologous regimens, with the addition, however, of a high level of antibody binding to an SD2 region spanning residues 616–627 and an HR2 region spanning residues 1160–1169. This group displayed a more focused antibody response with markedly lower omnipresent binding, and lower binding to mutational spots, although with a potent antibody response at the targeted regions. This diverging antibody binding profile was associated with a slightly improved ability to cross-neutralize variants (Appendix A).

A SARS-CoV-2 vaccine design using a native spike protein, as used for this study cohort, may induce a different RBD-binding antibody repertoire compared to that induced by a pre-fusion-stabilized spike protein. The pre-fusion-stabilized S-2P spike used in the mRNA-1273 vaccine [25] is believed to adopt a more open conformation with one or more monomers in the ‘up’ state in comparison with the native spike protein [3,26,27]. With a maximum RFU of 8529 (Beta-vaccinated group), none of the vaccinated groups tend to focus their antibody repertoire against the RBD. The antibodies in all three groups were mainly targeting class 4 and class 1 epitopes, both of which target RBD ‘up’ conformations. These epitopes are not accessible in the closed formation. Surprisingly, all three vaccinated groups targeted an epitope containing the glycan-shielded N343 with considerable intensity (Figure 2). The N343 RBD glycan is, besides shielding, suspected to interact with the ACE2 backbone and stabilize ACE2-RBD binding [4,28]. The mAb S309 (classified as a class 3 antibody) does not bind this exact epitope; however, N343 is sandwiched between the mAb’s binding sites, and the binding of this antibody possibly compromises ACE2-RBD binding stability, thereby inhibiting infectivity [28]. The observed antibody binding to the peptide containing N343 in all the groups of vaccinated mice may behave similarly. Despite the usage of a native spike, supposedly adopting a more ‘closed’ spike conformation compared to the pre-fusion-stabilized spike, the main antibody targeting is directed against epitopes only accessible in the ‘open’ conformation. Antibodies directed against the down state are regarded as being more potent, and, because of a more conserved area of the exposed surface of the closed conformation, they are also believed to have a higher level of cross-neutralization [10,27]. The observed weak antibody binding to closed conformation epitopes, combined with overall moderate RBD targeting, point towards effective antibody repertoires targeting regions outside the receptor-binding domain. 

As opposed to moderate RBD antibody targeting, we observed high antibody binding just outside the RBD, in the SD1 region, most noticeably observed in the Index-vaccinated group. The non-mutated sequence of peptide 579 displayed high antibody binding even at a 1:100 dilution compared to the remaining targeted peptides in this group, indicating a higher proportion of high-affinity antibodies targeting this peptide (Appendix A). For serum diluted 1:20, this peptide had an RFU of 12,733. However, a single substitution, A570D, significantly increased the antibody binding to the neighboring mutated peptide spanning residues 569–578 (IDDTTDAVRD) to an RFU of 36,706 (3-fold). The A570D substitution has been reported to increase viral entry into target cells [29]. The targeted peptide is overlapping with a known broadly neutralizing epitope in the SD1 [8,30,31,32,33] that presumably functions as a hinge of the open/closed state of RBD. Vaccination with the native spike, used for this study cohort [16], is known to elicit antibodies targeting this region [34]. In contrast, this SD1 region is occluded when using a vaccine design with a pre-fusion-stabilized spike protein. Being immunized with the Index strain without the A570D substitution, it is surprising that the elicited antibodies bind the peptide with the substitution with 3-fold greater intensity. However, despite high-resolution amino acid coverage on the microarray, an amino acid shift of two amino acids could have some limitations. Thus, an exact comparison of an identical peptide frame differentiating only in the substitution was not possible. The enhanced antibody fit could therefore partly be explained by the slight frame shift varying two terminal amino acids. It is unlikely, however, to be the sole reason for the increased signal as the terminal amino acids are positioned close to the linker attaching the 10′mer circular peptide to the glass plate; hence, the minimal epitope is likely unaffected by these amino acid changes. 

The S1 subdomain, and in particular the RBD, has been the primary target for research within neutralizing antibodies. However, the more conserved regions of S1 and S2 are speculated to have a higher contribution to cross-reactivity [35,36]. Residues 616–627 of the S2 region are parts of the 630 loop stretching from residue 617 to residue 644 [37]. The 630 loop is, along with the fusion peptide, believed to play a critical role in stabilizing the spike protein and serves a function in retaining the RBD in the ‘down’ mode [37,38,39]. Targeting this region with antibodies could possibly disturb the structural rearrangements necessary for opening up the RBD, leaving it in a locked ‘down’ position. The stability of the 630 loop is affected by the neighboring H655Y substitution; however, all three vaccinated groups share enhanced binding to the mutated form (Figure 3) [37]. HR2, which interacts with HR1 in the formation of the 6-helix bundle during membrane fusion [39,40], similarly reveals a broadly neutralizing epitope that becomes accessible after ACE2 binding [40,41,42]. Heavily targeting of this epitope, as observed in the mixed-vaccinated group, is likely to interfere with membrane fusion. Binding to HR2 has been reported to correlate with disease severity of SARS-CoV-2 [43].

## 5. Conclusions

Our results indicate that a heterologous vaccine regimen does not simply elicit an antibody response targeting the sum of antibodies induced by each individual vaccine. In contrast, a broader focused regimen induces an immune response tweaked to fit regions that could serve as common denominators, increasing neutralization breadth. Vaccinating with different variants might direct the immune system into a conserved region-directed response, targeting areas serving critical functions that are less likely to mutate and are evolutionarily “fixed”. Including a third vaccine, based on an additional variant such as the Omicron variant, or reversing the order of vaccination priming with the Beta vaccine and boosting with the Index vaccine, could broaden our understanding of this phenomenon. Furthermore, combining a heterologous vaccination regimen with a stepwise functional antibody characterization could provide interesting insight into the transition towards a conserved region-directed immune response. 

## Figures and Tables

**Figure 1 vaccines-12-00218-f001:**
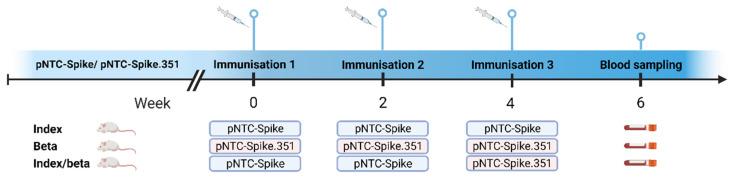
Vaccination regimens. Animals were immunized with either a pNTC DNA vector containing the full SARS-CoV-2 Index strain spike protein (pNTC-Spike) or the full SARS-CoV-2 Beta variant (pNTC-Spike.351) at two-week intervals. Immunizations were carried out according to the indicated 3 different regimens: Index, Beta, and mixed. Mice were immunized intradermally with a needle at the base of the tail, with a 50 µg dose. Blood samples were taken 2 weeks post final immunization. Vaccination regimens are further described in Lassauniere et al., 2021 [16].

**Figure 2 vaccines-12-00218-f002:**
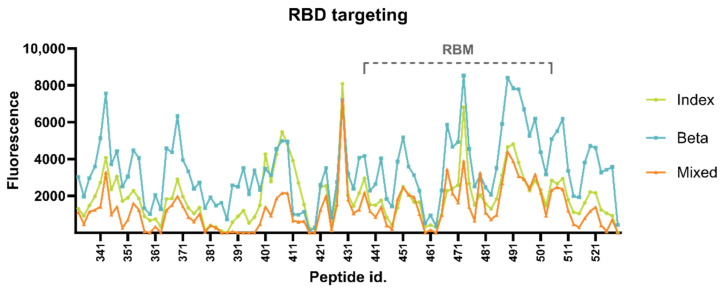
Antibody binding to the receptor-binding domain (RBD). Peptide id. on the *x*-axis refers to the position of the last amino acid of each 10′mer peptide on the microarray. The *y*-axis indicates the level of antibody binding to each peptide measured as the average of duplicate determinations of spot fluorescence minus background fluorescence normalized according to a positive control included on each subarray. RBM = receptor-binding motif.

**Figure 3 vaccines-12-00218-f003:**
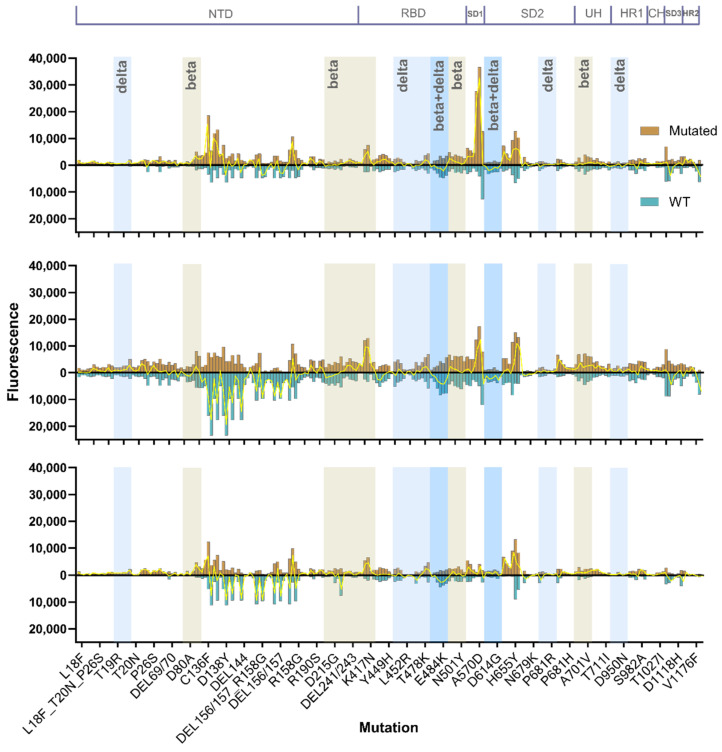
Comparative antibody binding to the non-mutated Index strain peptides mirrored to the mutated counterpart. The *x*-axis indicates the mutation; on the third peptide for each variant, the residue of interest is approximately in the center of the sequence. Substitutions make up 6 individual peptides, deletions make up 5 peptides, and deletions and substitutions make up 6 peptides. The *y*-axis indicates the level of antibody binding to each peptide measured as the average of duplicate determinations of spot fluorescence minus background fluorescence normalized according to a positive control included on each subarray. Values in gold indicate antibody binding to the mutated sequence and values in turquoise antibody binding to Index strain sequence. Yellow lines represent the antibody binding to the mutated peptide minus antibody binding to the corresponding Index strain peptide. Beta and Delta mutations are marked in colored bars.

**Figure 4 vaccines-12-00218-f004:**
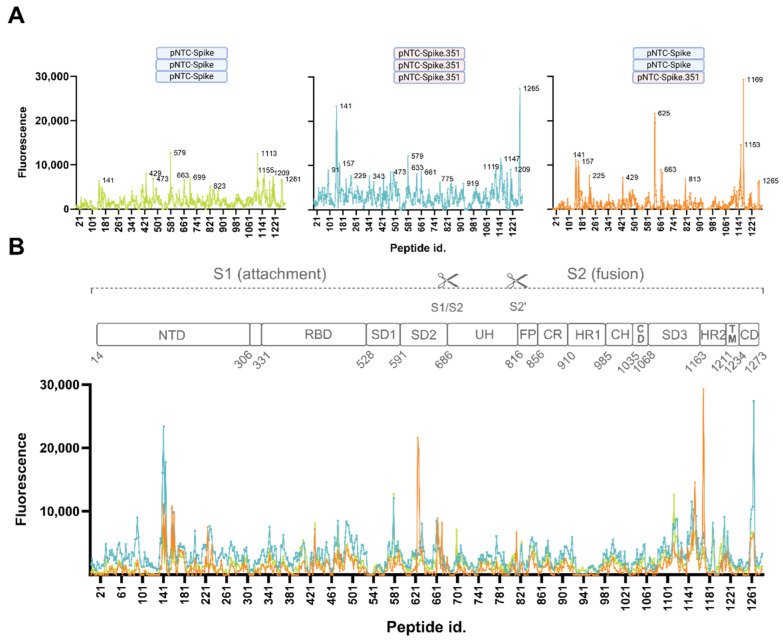
Serum-binding antibody profiles. Antibody binding to 10′mer overlapping peptides covering the entire SARS-CoV-2 Index strain spike ectodomain. The *x*-axis refers to the position of the last amino acid of each 10′mer peptide on the microarray. The *y*-axis (**A**,**B**) indicates the level of antibody binding to each peptide measured as the average of duplicate determinations of spot fluorescence minus background fluorescence normalized according to a positive control included on each subarray. (**A**) Individual binding profiles from antibody binding in pools of serum samples stratified according to vaccination regimen. Major peaks are labeled. (**B**) Full overview of antibody binding in all three groups relative to the different SARS-CoV-2 spike protein domains. S1 subunit (residues 14–685); S2 subunit (residues 686–1273); NTD = N-terminal domain; RBD = receptor-binding domain; SD1, SD2, SD3 = sub-domain 1, 2, 3; S1/S2 = furin cleavage site; UH = upstream helix; S2’ cleavage site; FP = fusion peptide; CR = connecting region; HR1 and HR2 = heptad repeat sequence 1 and 2; CH = central helix; CD = connector domain; TM = transmembrane helix; CD = connector domain. (**C**) Heatmap showing serum antibody binding relative to the different SARS-CoV-2 spike protein domains in the Index-strain-vaccinated group, the Beta-VOC-vaccinated group, and the mixed-vaccinated group.

## Data Availability

The raw data supporting the conclusions of this article will be made available by the authors on request.

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
