# Peer review of "Antigen-Heterologous Vaccination Regimen Triggers Alternate Antibody Targeting in SARS-CoV-2-DNA-Vaccinated Mice"

_vaccines, 2024, doi:10.3390/vaccines12030218_

Round 1
Reviewer 1 Report
Comments and Suggestions for Authors
The manuscript entitled: “Antigen heterologous vaccination regimen triggers alternate an-2 tibody targeting in SARS-CoV-2 DNA vaccinated mice” by Frische et al. deal with a very interesting topic - analysis of antibody epitopes following DNA vaccination against SARS-CoV-2. This study is significant since SARS-CoV-2 is still affecting millions of people worldwide. It is important to have more knowledge for the future vaccine research. However, there are still points to improve this article. Please see the comments below.
Abstract:
According to https://www.mdpi.com/authors/layout, the abstract contains a summary of the entire paper and can be up to 200 words long with only one paragraph. Yours abstract is a bit longer.
Introduction:
Line 89.
Change please by Lassauniere et al 2023 to by Lassauniere et al. [15]
Materials and methods:
· The DNA vaccines should be described in more details.
· For some common time units and measurement units, it is recommended to use abbreviated units if Arabic numerals are in front of them.
Lines 113-114
The third group received a combination of the two vaccines as the pNTC-Spike vaccine administered at weeks 0 and 2, and the pNTC-Spike vaccine.351 administered at week 4. Why the fourth group was not used with two injections of pNTC-Spike.351 and one pNTC-Spike?
Line 136
Pepperprint.com Missed city, country
Line 150
Thermo Fisher Missed city, country
Results:
Please, rewrite this section. Remove discussion elements from the Results, put them into Discussion.
Lines 197-198
The amount of individual mouse serum in each pool was normalized according to previously established levels of ELISA binding IgG titers. It would be nice to see these data here.
Discussion:
Discuss why Index strain and Beta variant of SARS-Cov2 were used. Have you compared sequence of obtained epitopes with spikes from other VOCs?
Line 329
Dot is missed
References:
Adapt the way of describing references to the magazine's standards.
General comment:
Pease reformat the paper according to https://www.mdpi.com/authors/layout. Remove extra lines between paragraphs, pay attention on references etc.
Author Response
Responses to Reviewers Comments
Independent Review Report, Reviewer 1
Response: We would like to thank the Reviewer for their time and constructive comments.
Abstract:
According to https://www.mdpi.com/authors/layout, the abstract contains a summary of the entire paper and can be up to 200 words long with only one paragraph. Yours abstract is a bit longer.
Response: We thank the reviewer for noticing, and we have reduced the Abstract to 199 words.
Introduction:
Line 89.
Change please by Lassauniere et al 2023 to by Lassauniere et al. [15]
Response: Thank you for noticing. We have changed the reference in the text accordingly.
Revised manuscript, Page 2, line 80-83: ‘In studies conducted by Lassauniere et al. [15], where CB6F1 mice received a homologous regimen of three immunizations of either of these vaccines or a combination of both, the authors observed an increased breadth of neutralization responses when the vaccines were mixed.’
Materials and methods:
- The DNA vaccines should be described in more details.
Response: We agree with the reviewer that the manuscript would benefit from additional details about the vaccines used for immunizing the animals in this study cohort. We have revised the manuscript accordingly.
Revised manuscript, Page 2-3, line 91-101: ‘The DNA vaccines, vaccination strategies, and animal experiments are described in detail elsewhere [15,16]. In brief, eight-week-old female CB6F1 mice (Envigo, Netherlands) were immunized with 50 µg of a DNA plasmid vector, encoding either the complete and unaltered spike protein of the SARS-CoV-2 Index strain (pNTC-Spike), derived from the Wuhan-Hu-1 strain (MN908947), or of the Beta (B.1.351) strain (pNTC-Spike.351). Nature Technology Corporation (Lincoln, NE, USA) subcloned human codon-optimized SARS-CoV-2 spike sequences synthesized by GeneArt (Thermo Fisher Scientific, Dreieich, Germany) into the NTC8685-eRNA41H vector backbone and produced the vaccines in NTC4862 E. coli cells (DH5α attλ::P5/6 6/6-RNA-IN-SacV, Cmr) using their antibiotic-free RNA-OUT selection procedure. The vaccine stocks were provided in a concentration of 10 mg/mL in phosphate buffered saline (PBS). [15].’
For some common time units and measurement units, it is recommended to use abbreviated units if Arabic numerals are in front of them.
Response: We have updated the text throughout the manuscript with abbreviations of minutes, seconds etc.
Lines 113-114
The third group received a combination of the two vaccines as the pNTC-Spike vaccine administered at weeks 0 and 2, and the pNTC-Spike vaccine.351 administered at week 4. Why the fourth group was not used with two injections of pNTC-Spike.351 and one pNTC-Spike?
Response: We agree with the reviewer that inclusion of a fourth vaccinated group vaccinated in the reverse manner with a Beta prime and Index boost could have been an interesting addition to the study. At the time the mouse study was designed, the intention was to inform preparation for a phase I clinical trial for the Beta vaccine as a variant update vaccine in the pandemic and then it would have been administered to people who already received a vaccine with the index strain. This is the reason for the index-index-beta order. There was no real anticipation that, in the real world, people would receive the Beta vaccine and subsequently receive an index vaccine. For these reasons, this regimen was not tested.
We have added a segment to the newly inserted section ‘Conclusions’ addressing this matter.
Revised manuscript, Page 11, line 412-423:
‘5. Conclusions
‘Our results indicate that a heterologous vaccine regimen does not simply elicit an antibody response targeting the sum of antibodies induced by each individual vaccine. In contrast, a broader focused regimen induces an immune response tweaked to fit regions that could serve as common denominators increasing neutralization breadth. Vaccinating with different variants might direct the immune system into a conserved region-directed response, targeting areas serving critical functions that are less likely to mutate and are evolutionary “fixed”. Including a third vaccine, based on an additional variant such as the Omicron variant, or reversing the order of vaccination priming with the Beta vaccine and boosting with the Index vaccine, could broaden our understanding of this phenomena. Furthermore, combining a heterologous vaccination regimen with a stepwise functional antibody characterization, could provide interesting insight into the transition towards a conserved region-directed immune response.’
Line 136
Pepperprint.com Missed city, country
Line 150
Thermo Fisher Missed city, country
Response: We have added these details to the manuscript.
Results:
Please, rewrite this section. Remove discussion elements from the Results, put them into Discussion.
Lines 197-198
Response: We thank the reviewer for the useful comment improving the quality of the manuscript. We have rewritten the Results section, and moved discussion elements to the Discussion or the Introduction for an improved flow of the text.
Revised manuscript, Page 10 (Discussion), line 349-357: ‘A SARS-CoV-2 vaccine design using a native spike protein, as used for this study cohort, may induce a different RBD-binding antibody repertoire compared to that induced by a pre-fusion stabilized spike protein. The prefusion stabilized S-2P spike used in the mRNA-1273 vaccine [25] is believed to adapt a more open conformation with one or more monomers in the ‘up’ state in comparison with the native spike protein [3,26,27]. With a maximum RFU of 8529 (Beta vaccinated group), none of the vaccinated groups tend to focus their antibody repertoire against the RBD. Antibodies in all three groups were mainly targeting class 4 and class 1 epitopes, both of which target RBD ‘up’ conformations.’
The amount of individual mouse serum in each pool was normalized according to previously established levels of ELISA binding IgG titers. It would be nice to see these data here.
Response: We have added the ELISA data to Supplementary Material, Sup Fig 1C.
Discussion:
Discuss why Index strain and Beta variant of SARS-Cov2 were used. Have you compared sequence of obtained epitopes with spikes from other VOCs?
Response: We have added a small section to the ‘Discussion’ aiming to clarify the usage of the Beta VOC in the vaccine as an alternative to the ancestral Index strain. We have not compared sequence of obtained epitopes with spikes from other VOCs.
Revised manuscript, Page 9 (Discussion), line 319-330: ‘It is a follow-up study of the development and pre-clinical immunogenicity studies of the SARS-CoV-2 DNA vaccines described in Lassauniere et al. [15,16]. These studies found an enhanced breadth in heterologous vaccinated mice that received a combination of a DNA plasmid encoding the full Index strain spike protein sequence and an identical plasmid vector encoding the full Beta VOC spike protein. Being one of the first variants bearing immune escape capabilities such as the E484K and K417N/T substitutions, the Beta VOC represented a highly antibody resistant variant compared to the ancestral Index strain. To determine if the observed improved ability to cross-neutralize different VOCs in the heterologous vaccinated mice, is the result of a different antibody targeting, we established B-cell epitope profiles from polyclonal serum from groups of mice representing each vaccination regimen.’
Line 329
Dot is missed
Response: Dot has been added.
References:
Adapt the way of describing references to the magazine's standards.
General comment:
Pease reformat the paper according to https://www.mdpi.com/authors/layout. Remove extra lines between paragraphs, pay attention on references etc.
Response: We thank the reviewer for noticing format issues not complying with MDPI standards. We have updated the format, removed extra lines between paragraphs, inserted a ‘Conclusions’ section, and updated Reference management to a format recently accepted by MDPI. We now hope that the layout fulfills the standards of the Journal.
Reviewer 2 Report
Comments and Suggestions for Authors
In this study, the authors used microarray method to analyze the antibody epitopes targeted during the heterologous vaccination in mice and discovered broad neutralizing antibody response. This study provides valuable information in light of ever mutating virus and a need for developing a broad-spectrum SARS-CoV-2 vaccine. The research design is appropriate and the manuscript is very well written.
Minor comments:
1. In Figure 3, gold indicates mutated and turquoise indicates WT. But in the Figure 3 legend, it’s written- “Values in gold indicate antibody binding to the Index strain sequence and values in turquoise antibody binding to the mutated sequence.” This needs to be corrected.
2. Figure 4C: Describe the results in detail.
Comments on the Quality of English LanguageThis study provides valuable information in light of ever mutating virus and a need for developing a broad-spectrum SARS-CoV-2 vaccine. The research design is appropriate and the manuscript is very well written.
This manuscript can be accepted after the authors address minor comments.
Author Response
Responses to Reviewers Comments
Independent Review Report, Reviewer 2
Response: We would like to thank the Reviewer for their time and constructive comments.
In this study, the authors used microarray method to analyze the antibody epitopes targeted during the heterologous vaccination in mice and discovered broad neutralizing antibody response. This study provides valuable information in light of ever mutating virus and a need for developing a broad-spectrum SARS-CoV-2 vaccine. The research design is appropriate and the manuscript is very well written.
Minor comments:
- In Figure 3, gold indicates mutated and turquoise indicates WT. But in the Figure 3 legend, it’s written- “Values in gold indicate antibody binding to the Index strain sequence and values in turquoise antibody binding to the mutated sequence.” This needs to be corrected.
Response: We thank the reviewer for noticing this mistake and we have corrected the manuscript accordingly.
Revised manuscript, Page 7, line 252-258: ‘The y-axis indicates the level of antibody binding to each peptide measured as the average of duplicate determinations of spot fluorescence minus background fluorescence normalized according to a positive control included on each subarray. Values in gold indicate antibody binding to the mutated sequence and values in turquoise antibody binding to Index strain sequence. Yellow lines represent the antibody binding to the mutated peptide minus antibody binding to the corresponding Index strain peptide.’
- Figure 4C: Describe the results in detail.
Response: Figure 4C is intended to support conclusions drawn from Figure 4A and B, however with an improved overview of each groups ability to target their response on individual epitopes. We have added references within the section to Figure 4C including a sentence describing differences in antibody focus observed especially in the Beta vaccinated group and the mixed vaccinated group.
Revised manuscript, Page 9, line 308-318: ‘Peptide 1169 (TSPDVDLGDI) in the HR2 region similarly showed very high antibody binding in the mixed vaccinated group (RFU = 29,320) compared to 4283 and 6584 RFU as seen in the Index and in the Beta vaccinated group, respectively. The Beta vaccinated group showed high general antibody levels covering most parts of the spike protein and focusing less on individual epitopes, in contrast to the more focused response observed for the mixed vaccinated group (Fig 4C). Serum antibodies of the mixed vaccinated group does however seem to target regions in the NTD, RBD, CR, and CH also observed for the Beta vaccinated group, which were absent to low in the Index vaccinated group (Fig 4C). Of note, none of the groups displayed noticeable high levels of binding to the S1/S2 cleavage site, the S2’cleavage site nor the fusion peptide.’
Comments on the Quality of English Language
This study provides valuable information in light of ever mutating virus and a need for developing a broad-spectrum SARS-CoV-2 vaccine. The research design is appropriate and the manuscript is very well written.
This manuscript can be accepted after the authors address minor comments.